# Position: RL Researchers Need to Distinguish Between Solving Simulators and Using Simulators as a Proxy

**Matthew Vandergrift** [1 2]  **Esraa Elelimy** [1 2]  **Martha White** [1 2 3]

## Abstract

One goal in reinforcement learning (RL) research is to understand general-purpose sequential decision-making, using benchmark simulators as a proxy for learning in deployment settings. When running experiments, however, the goal of achieving high performance in the simulator can mutate into focusing exclusively on solving the simulator. To achieve high scores, researchers may adopt solutions exclusively meant for solving simulators, rather than learning while the agent is deployed outside a simulator. Solving simulators is also worthy of investigation, but it is a fundamentally different RL research question. *In this paper, we argue that RL researchers need to distinguish between two use cases of simulators: solving simulators and using simulators as a proxy for learning in deployment.* We first discuss how these two use-cases are importantly different, in terms of constraints on how the agent can use the simulator, which algorithms are appropriate, and which evaluation metrics are appropriate. We then highlight several issues and misleading conclusions that can occur by not making the distinction between these two settings clear, supported with examples and simple experiments. This work is a call to the community to begin clearly distinguishing how they are using simulators in their work, hopefully sparking further discussion on which empirical practices work best in each setting.

## 1. Introduction

Most RL papers do empirical comparison of algorithms in simulators. There are a host of typical research simulators, including classic control, MuJoCo (Todorov et al., 2012), Arcade Learning Environment (Bellemare et al., 2013), Behaviour Suite (Osband et al., 2020), Minecraft (Guss et al., 2019), and DeepMind-Control (Tunyasuvunakool et al., 2020). Other papers use RL algorithms in more realistic simulators, such as for a Tokomak reactor (Degrave et al., 2022b) or the advanced simulator tools from the process control community, such as Aspen HYSYS or DWSim, for problems like carbon capture (Al-Sakkaria et al., 2024) and water treatment (Gao et al., 2023).

The ubiquitous use of simulators can mask two very different goals in papers: solving simulators or using simulators as a proxy for learning in deployment. For the first group, the primary goal is to solve a specific simulator. For example, an engineer may have a simulation of a wastewater treatment plant they intend to build, and would like to identify an optimal controller to assess the design. As another example, a games researcher may want to find the best Go player, as was done by AlphaGo (Silver et al., 2016). Critical research questions can be answered through solving simulators. For instance, whether Strassen's two-level algorithm was optimal for $4 \times 4$ matrix multiplication was answered by AlphaTensor (Fawzi et al., 2022), and protein docking molecules were discovered by solving a protein docking simulation (Kim et al., 2024).

For the second group, the simulator is only used as a proxy for learning in deployment. We define the deployment setting as the setting where the dynamics of the environment are outside the control of the practitioner, as is the case when controlling physical systems or interacting with people. Additionally, for learning in deployment, the agent interacts with this single deployment environment, and rewards during that interaction matter. For many researchers in this setting, it would be preferable to run all experiments in deployment. For example, to truly test whether a new RL algorithm can reduce energy costs in a data center, it would be ideal to deploy it directly in a data center and observe any reduction in energy use as it learns and acts on the physical system. But, naturally, running such an experiment is much more onerous. For algorithm development, research simulators allow investigating new ideas quickly and cheaply, and allow running controlled experi-

[1]Department of Computing Science, University of Alberta [2]Alberta Machine Intelligence Institute (Amii) [3]Canada CIFAR AI Chair. Correspondence to: Matthew Vandergrift <mwvander@ualberta.ca>.

*Proceedings of the 43ʳᵈ International Conference on Machine Learning*, Seoul, South Korea. PMLR 306, 2026. Copyright 2026 by the author(s).

ments. Some brave researchers have taken the next step of also running RL experiments in real deployment scenarios, such as in robotics (Büchler et al., 2022; Liu et al., 2025; White et al., 2012; Cheng et al., 2019; Ma et al., 2023), recommender systems (Bietti et al., 2021), and for data center cooling (Luo et al., 2022). But the vast majority of RL researchers stick to experiments in simulation.

When doing experiments in simulation rather than in deployment, it is easy to start inadvertently cheating and exploiting the fact that an algorithm is being tested on a simulator. Thus, a researcher must be hyper-vigilant to constrain the use of the research simulator, to ensure it remains a reasonable proxy for deployment scenarios. When running experiments, however, the short-term goal can morph into solving the research simulator to get good benchmark scores, even though the long-term goal is to understand the algorithms faithfully in simulation, ultimately for practical use. Due to the pressures from benchmarking, it is no surprise that the line between these two goals can become murky.

As an anecdotal example, consider the introduction of learning in parallel in Atari with A2C (Mnih et al., 2016). This method is implicitly designed for the problem of solving simulators, because we cannot have multiple parallel copies of the environment in deployment settings.[1] Yet, the A2C paper did not explicitly state this as its goal and instead stated a goal of simply having a better general RL agent. Many following methods built on this innovation, such as vectorized PPO (Schulman et al., 2017), IMPALA (Espeholt et al., 2018), PQN (Gallici et al., 2025), because it is a great way to solve a simulator more quickly. These papers similarly did not state that their methods were restricted to solving simulators. The ensuing arms race to improve performance in Atari led to Go-Explore (Ecoffet et al., 2021), which further exploited access to the simulator by resetting the agent to different states for faster learning. There was some negative reaction to this paper, because such resetting was "cheating", again indicating confusion in the community around these goals. If the Go-Explore paper had stated a goal of solving simulators (getting state-of-the-art on Atari in this case), then resetting is not cheating, and such a negative reaction should not occur. Of course, outrage is a minor consequence; in this paper, we outline that there are many more serious issues with the lack of clarity between these two goals in the RL community.

---

[1]There can be instances where deployment involves something resembling parallelism, for instance, a recommendation algorithm can asynchronously provide recommendations to many users. In this case, however, one simulated environment would be the collection of users, i.e., it would include that type of parallelism, whereas parallelizing a simulator would correspond to increasing the number of recommendation platforms. In this work, we define our deployment setting to mean that the agent interacts in a single environment.

In this paper, we take the position that **RL researchers need to distinguish between solving simulators and using simulators as a proxy for learning in deployment**. We first outline this distinction in terms of differing problem formulations, highlighting the ramifications on agent-environment interaction, algorithm development, and evaluation metrics. We then dive deeply into four specific issues that arise from not distinguishing between simulator use cases and provide a call-to-action for addressing these issues. We conclude with a discussion on alternative views. Our work joins others outlining misalignments in empirical practices in reinforcement learning (Patterson et al., 2024; Castro, 2025).

Note that solving a simulator and using a simulator as a proxy for learning in deployment are not the only two goals in RL. There are a variety of problem settings that may not neatly fall into these two categories. In this position paper, however, we focus on these two use cases—solving a simulator and learning in deployment—to keep a manageable scope. Additionally there may be research problems which apply to both of these goals. As an example, research towards better optimizers for RL might produce advances which benefit both solving simulators and learning in deployment. In general there can be many works which produce conclusions that apply to both settings, however this does not imply that this will always be the case, as we will show and discuss throughout this work.

**Remark about sim2real:** It is worth explicitly noting that the simulation-to-reality, known as sim2real (Zhao et al., 2020; Salvato et al., 2021), use case is different. The goal is to learn a policy in a high-fidelity simulator of the real world task, and then deploy that policy in the real world task. There is no confusion in this setting on the role of the simulator, and research questions often revolve around the sim2real gap and improving the simulator to produce more effective policies for the real world. A substep of sim2real is to solve a high-fidelity simulator, and so algorithms for the solving simulators use case could be useful for sim2real. Using a research simulator as a proxy is arguably more distinct, in that the goal is to obtain empirical insights in the research simulator rather than solve a specific engineering problem or application. Developing better algorithms by using research simulators can of course be beneficial for sim2real, particularly in the step in sim2real where the policy is fine-tuned or updated once deployed in the real world. Nonetheless, neither of our two use cases are explicitly about sim2real, and to maintain a manageable scope, we avoid including the related but distinct use case of sim2real.

## 2. Simulator Usage Determines Problem Formulation

How a simulator is being used has implications on agent-environment interaction constraints, applicable solution

*Table 1.* The distinguishing criteria between our two simulator use cases. The bold in the rightmost column denotes the better choice for the algorithms for solving a simulator, because it exploits parallelism with multiple environments and/or simulator-exclusive actions.

|  | Using Simulator as a Proxy | Solving a Simulator |
| --- | --- | --- |
| Interaction Options | 1 environment | $n$ environments (can use $n = 1$) 
 Simulator-exclusive actions: 
 (1) resetting to a specific state and 
 (2) simulating multiple outcomes |
| Solution Techniques | Model-free algorithms (e.g., Sarsa($\lambda$) (Sutton & Barto, 2018), DQN (Mnih et al., 2015), PPO (Schulman et al., 2017)) 

 Planning with a learned model (e.g., MuZero(Schrittwieser et al., 2019)) 

 Meta-learning algorithms for setting hyperparameters | Model-free algorithms 
 **Multi-stream model-free algorithms** 
 **(e.g., PPO, A2C, A3C (Mnih et al., 2016), PQN (Gallici et al., 2025))** 
 Planning with a learned model 
 **Planning with a simulator** 
 **(e.g., Dynamic Programming, AlphaZero (Silver et al., 2018))** 
 Meta-learning algorithms for setting hyperparameters 
 **Parallel Tuning (e.g., grid search)** 
 **Resetting for exploration (e.g., Go-Explore (Ecoffet et al., 2021))** 
 **Resetting for variance-reduction (e.g., vine method TRPO (Schulman et al., 2015))** |
| Evaluation Metrics | Average episodic return over learning 

 Sample efficiency | Best return: expected return for the best policy found during learning 
 Total computation used |

techniques, and evaluation metrics. In this section we begin by formally defining a simulator, then we outline these implications for solving a simulator and for using a simulator as a proxy for learning in deployment. These implications, listed in Table 1, showcase a clear dichotomy between the two simulator usages we discuss.

Before we dive into our problem settings, we provide a definition for the term *simulator* as we use it in our work. A simulator is a software implementation of an environment dynamics in which interactions are consequence-free: transitions and rewards are recorded but not physically enacted, so that no real-world costs, risks, or irreversible effects are incurred. Furthermore, since actions are consequence-free, some simulator-exclusive actions are possible, such as parallel copies of the simulator, resetting to arbitrary states, and querying multiple outcomes from the same state-action pair. We expand upon this definition throughout the rest of this section by discussing how simulators can be used in the two problem settings of interest.

There are two key distinguishing interaction constraints between these two settings: the ability to use parallel environments and the availability of simulator-exclusive actions. When solving a simulator, it can be feasible to receive asynchronous observations from parallel copies of a simulator. For RL in a deployment setting, observations occur sequen-

tially from the environment. When emulating this setup with a simulator—using it as a proxy—a single copy of the simulator should be used.

Simulator-exclusive actions revolve around the ability to directly query the simulator from a chosen state. When solving a simulator, the agent can reset to a specific state, potentially to facilitate exploration and learn more about an area. It can also query multiple outcomes, given a particular state and action. We can think of resetting and resampling as simulator-exclusive actions because they are outside the typical agent-environment interaction loop and only available because the practitioner has control over the simulator. In deployment, an agent cannot directly resample or reset since the practitioner does not have complete control over the environment.[2] If these simulator-exclusive actions are used, then the experiment in simulation is not a faithful proxy.

The differing constraints on interaction result in different solutions being appropriate. Any algorithm designed for learning in deployment—learning sequentially from a single

---

[2]This action can potentially be mimicked by giving the agent an explicit reset action that requests a human move the agent, as can be done in robotics. However, such an action is now part of the environment, rather than an action outside the environment, and is not the same as these simulator-exclusive actions.

stream of interaction—can be applied to solving a simulator. However, such algorithms leave a lot of performance on the table, since they do not take advantage of parallelism and simulator-exclusive actions. Once an algorithm leverages these two options, though, it is no longer applicable for learning in deployment. This is reflected in Table 1, where the algorithms in the left column for simulators-as-a-proxy are repeated in the right column for solving a simulator, with the additional options given for the right column exploiting these two additional options. Currently, RL algorithms are rarely proposed with explicit distinctions between these use cases. As evidence, vectorizing PPO is a common practice, but PPO with simulator resets is not (Huang et al., 2022).

The goals for solving a simulator and learning in deployment are often different, and so the evaluation metrics are also different. When the goal is to solve a simulator, the typical goal is to find an optimal policy. The agent can periodically pause learning and compute rollouts of the current (greedy) policy to get an estimate of the expected return and store this policy if it is the best-to-date. At the end of learning, it can return the best policy it has found so far. This metric is either reported as the *best episodic return*—with the correct but longer title of expected episodic return of the best policy found—or as the *optimality gap*.

For learning in deployment, on the other hand, a typical goal is to obtain as much reward as possible during learning. This goal is the source of the exploitation-exploration dilemma, where the agent has to balance between taking an action it believes gives more reward versus one to learn more about the environment. If the goal was just to reach an optimal policy, rather than to maximize reward along the way, then the agent could be fully exploratory, as long as it found the optimal policy faster. Since the rewards during learning do matter, it is typical to use average episodic returns or average reward across all steps of learning (Patterson et al., 2024). A common way to summarize all rewards during learning is to report the *area under the learning curve (AUC)*.

The distinction can be made concrete by considering the learning curve shown in Figure 1. The figure shows the return from a saved checkpoint of the policy that had the highest return before the dashed line, compared to the online return. This result is interpreted very differently depending on the use case. The result is good when solving a simulator, because the best policy found can be saved and reused, achieving similar performance to many state-of-the art RL algorithms (Achiam, 2018). In contrast, when using the simulator as a proxy, this performance is bad, as the agent's performance deteriorates over time. The agent cannot learn continually, which is an essential trait for agents that learn through interaction in a deployment environment. We provide additional details about this experiment in Appendix A.1.

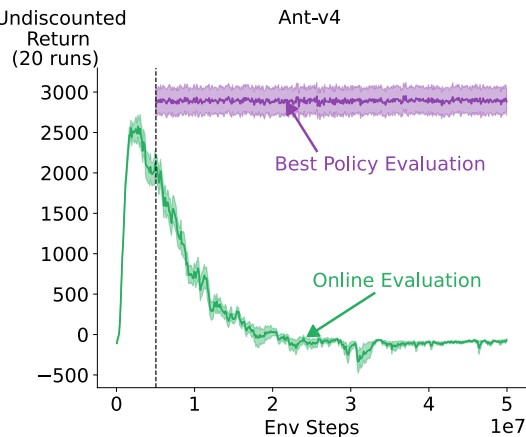

*Figure 1.* Example of a learning curve that presents a reasonable solution when solving a simulator but presents a bad solution when using simulators as a proxy. The solid line is the average online performance across 20 runs and the shaded area is the standard error.

There is a lot of room for nuance in this evaluation of learning in deployment, given different use cases. It can be reasonable to focus on performance in the second-half of learning if the use case allows an initial learning phase with no cost. For example, there might be a phase in a factory where an engineer sets up a robot and can babysit it during early learning. After this initial phase, the robot is evaluated while it is taking actions in the factory. It is hard to give one right choice for the simulator-as-a-proxy setting, because there are so many varied proxy scenarios. However, it is likely that evaluations will include measures of sample efficiency, performance during learning, and stability. Like solving simulators, it is also useful to compare compute costs and factor in compute restrictions for the intended deployment scenarios (e.g., edge devices versus powerful desktops). Despite this variety, the evaluation is still likely to be different from those used for solving a simulator.

## 3. Misleading Conclusions from Not Distinguishing Between Simulator Usages

In this section, we dive deeper into several differing criteria between solving a simulator and using a simulator as a proxy, highlighting issues when blurring the line between these two use cases.

### 3.1. Issue: Parallel Environments for the Simulator-as-a-Proxy setting

Despite their persistent appearance in science fiction, parallel worlds have yet to be discovered in the real world. When learning in deployment, the agent has access to the single deployment environment and cannot spin up additional copies to get more data. When using simulators as proxies for deployment, RL researchers should restrict their agents

from using parallel copies of the simulator. Learning in the simulator should aim to mimic the dynamics of learning in such a deployment setting. Understanding behavior in the learning process itself is a key focus for this use case, to ultimately develop algorithms that work well in deployment. The researcher themselves can run multiple experiments in parallel (i.e., over different random seeds), but the agent within a run should not.

Yet, using parallel environments is alluring because experiments can run much more quickly. Most libraries provide multi-stream algorithms and parallelization on GPUs to achieve significant speed increases (Lu et al., 2022; Deep-Mind et al., 2020; Bonnet et al., 2024; Lange, 2022), with algorithms like PPO (Schulman et al., 2017), PQN (Gallici et al., 2025), and A2C (Mnih et al., 2016). For modern libraries, the parallelization is often accessible via a single line of code, rather than the previously required strenuous engineering. This has allowed modern algorithms to implement the number of environments as a hyperparameter to be configured (Gallici et al., 2025; Schulman et al., 2017), rather than a fundamental change in the algorithm. Such multi-stream algorithms should be used when the goal is to solve a simulator, as it can significantly reduce the time to find an optimal policy, which is a key metric for that setting.

Using these same multi-stream algorithms for the simulator-as-a-proxy setting, however, can cause confusion and misleading results. Because RL papers do not always state their explicit goal between these two use cases, it may not always be clear to anyone inside or outside the community that such algorithms are suited to the solving-simulator setting rather than the simulator-as-a-proxy setting. This issue is particularly problematic because the results can look highly impressive—deep RL can learn complex policies—further enticing individuals and practitioners to consider using these multi-stream algorithms and their variants.

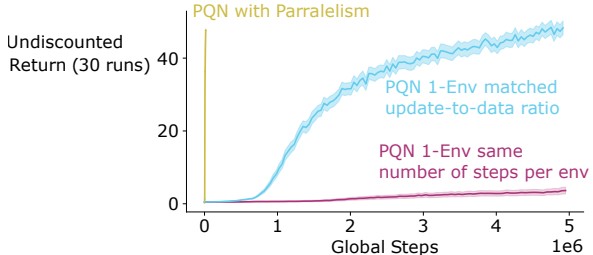

*Figure 2.* Comparing PQN with 128 parallel simulators to PQN with 1 simulator for Asterix-Minatar. A global step is one step in each available simulator, with the shaded region being standard error.

Suppose a practitioner wants to apply state-of-the-art RL algorithms to their domain, and that they have a domain-specific simulator to evaluate algorithms. This practitioner

might be compelled to use PQN due to its "competitive results in significantly less wall-clock time than existing state-of-the-art methods" (Gallici et al., 2025). To achieve this, PQN uses up to 1024 copies of a simulator, and is therefore only applicable for solving simulators. The paper introducing PQN does not make clear that it is intended for solving simulators, but it does provide the code to run PQN. The practitioner may not realize the restrictions for parallel environments and decide to test the available PQN implementation on their simulator. When evaluating PQN in their simulator, it will appear fast and performant. But when they need to shift to using one copy of the environment as a step towards deployment, this may not be the case.

We demonstrate this effect in Figure 2 using Asterix from MinAtar (Young & Tian, 2019) as a simulator. When switching to a single environment, PQN is much slower than with parallel environments. Switching to one environment and keeping the same number of updates lowers performance, and needs to resolved by matching the update-to-data ratio of the parallel environment version. This is done by increasing the number of rollout steps in the single environment. These results undercut the impressive speed which may have prompted this practitioner to use PQN in the first place. More details on this experiment are provided in Appendix A.2.

### 3.2. Issue: Suboptimal performance when not exploiting resetting

There are also consequences for solving simulators when not sufficiently separating these use cases, particularly in terms of leaving performance gains on the table. One way this manifests is that many algorithms that leverage parallel environments do not leverage resetting the simulator to specific states. This includes algorithms mentioned above, like PPO, A3C, PQN, and others. Yet resetting can be highly useful and has been explored in RL with generative models (Sidford et al., 2018; Lattimore et al., 2020; Azar et al., 2012). Mhammedi et al. (2024) showed that exploiting the simulator via resetting leads to provably better sample efficiency. Hao et al. (2022) developed a convergent algorithm with the assumption that it can reset to any state the agent has seen. Empirically, there has been impressive performance in the hard-exploration problems in Atari with algorithms like Go-Explore (Ecoffet et al., 2021) that leverages resetting.

Consider a practitioner attempting to solve a challenging simulator. They may choose to use PPO with parallel environments, a common implementation choice in PPO (Huang et al., 2022), to quickly expose the agent to more transitions. However, since resetting is not commonly implemented with PPO, they choose not to use it. This will likely limit the quality and speed of the learned policy in the simulator. If researchers were to distinguish between simulator usages,

then this in-between setup would not be common, leading to better results when applying RL algorithms.

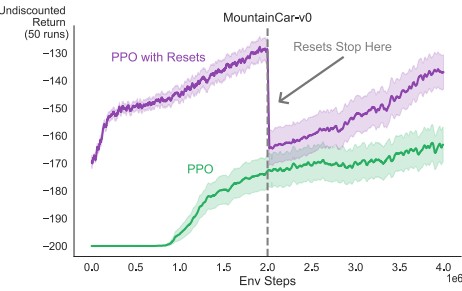

*Figure 3.* Demonstration of how resetting can lead to a better performing final policy for PPO compared to PPO without resets. Notably the policy maintains increased performance when evaluated without resets.

We can observe the performance left on the table by not using resets in an example simulator, MountainCar-v0 (Lange, 2022). For simplicity, we use a naive resetting mechanism, resetting the agent to a random state upon episode termination every 500 steps. After 2-million steps, we stop resetting. We stop resetting so that the agent can use the information gained by resetting to quickly learn a policy it can run in the environment. In Figure 3, we plot the online performance of the resetting and generic PPO. Figure 3 clearly shows that the resetting algorithm learns a better policy than the non-resetting one. The early resets provide the agent with useful experience such that without resets it quickly surpasses the non-resetting agent. These performance gains can be immensely valuable when solving a simulator of interest. Exact details about our experiment setup can be found in Appendix A.3.

### 3.3. Issue: Hyperparameter Tuning

Hyperparameter tuning presents a subtle opportunity for researchers to mistakenly shift their algorithm to be more appropriate for solving a simulator. No matter the problem, most practitioners would agree that hyperparameters are a nuisance, and RL algorithms notoriously contain many of them (Adkins et al., 2024). Moreover, hyperparameter values can have dramatic impacts on performance in RL (Duan et al., 2016). Researchers designing new algorithms or tackling new problems must decide how to set hyperparameter configurations before running experiments. The standard approach in the RL literature is to implement a sweep over hyperparameter values, guided by a search with varying degrees of intelligence (Eimer et al., 2023).

Hyperparameter tuning can exploit a simulator in two ways. Firstly, for each hyperparameter configuration tested, the simulator can be reset. As discussed in section 3.2, resetting is not viable when using the simulator as a proxy for learning in deployment. Secondly, when the best hyperparameter

configuration is chosen, all previous learning trials from the hyperparameter search can be thrown out. Throwing out results from the hyperparameter search exploits the simulator, as such an approach can not be done in deployment; an agent cannot try out a hyperparameter configuration without cost. In deployment, all rewards matter, and the consequences of choosing a bad or catastrophic hyperparameter configuration can not be erased. Erasing an environment and the associated costs incurred within it is only possible with simulators. Despite this exploitation of the simulator, it is currently standard to do this style of tuning even when the simulator is a proxy for learning in deployment.

Let us highlight this issue by considering a thought exercise, with a researcher proposing a novel RL algorithm for learning in deployment. Suppose they use the VizDoom simulator (Wydmuch et al., 2019) as a proxy for learning in deployment. The practitioner uses Rainbow as a baseline for comparison, due to its impressive results on the ALE benchmark (Hessel et al., 2018). Next, the practitioner will have to decide how to set Rainbow's 25 (Adkins et al., 2024) hyperparameters. This step is likely required, as it has been established that default hyperparameters are often not suitable for new problems (Jayawardana et al., 2022). They make the reasonable choice to follow what is reported in the Rainbow paper. They perform "manual coordinate descent" (Hessel et al., 2018), for the reported sensitive hyperparameters, within the provided ranges. To do this, they create a copy of VizDoom for each sensitive hyperparameter. In each copy, they test each value in the range by resetting the simulator and running for 1-million steps. Lastly, they choose the best value for each hyperparameter, start a new instance of VizDoom, and evaluate the performance of the configuration.

This is a faithful recreation of what is described in the Rainbow paper, but it is now solving the VizDoom simulator. Rainbow is not described as an algorithm built to solve simulators, nor does it use any simulator-exclusive actions. But the reliance on hyperparameter tuning inadvertently makes it more appropriate for solving simulators, and it becomes a less appropriate comparison to the researcher's algorithm, which is designed for learning in deployment.

### 3.4. Issue: Evaluation Rollouts

When showing learning curves in reinforcement learning experiments, it is common to use one of two metrics. The first is the *online return*. As the agent learns, the rewards it sees during learning are stored, and the online returns are calculated from these rewards. The other is an *evaluation rollout*, or offline return. Here, every few steps, the policy is frozen, and samples of returns are obtained by running this frozen policy in new instances of the environment. These sampled returns or rollouts are averaged to produce

an expected return for the policy at that learning step. The primary difference between these two is that the online return reflects performance *with exploration* for a learning agent, whereas the rollout reflects the quality of the policy found at a specific time, typically without exploration.

When using the simulator as a proxy for learning in deployment, online returns are typically a more sensible measure. To more concretely explain why, consider an RL agent controlling the chemical dosing for a water treatment plant. It is learning when to increase and decrease dosing while it controls the plant, and the amount of cost is incurs is given by the online return. The greedy policy at a given point is never deployed, and so the quality of this policy—given by an evaluation rollout—does not reflect the actual cost incurred.[3] It is possible that exploratory actions cause damage to the system, even if the greedy policy does not take these actions, and the online return would reflect this, whereas the evaluation rollout would not.

It is worth also pointing out that computing evaluation rollouts in deployment would require the agent to periodically stop learning and run a fixed policy to get returns. These rollouts should count as part of the interaction the agent has in deployment. Online returns, on the other hand, are just computed from the rewards during the learning process.

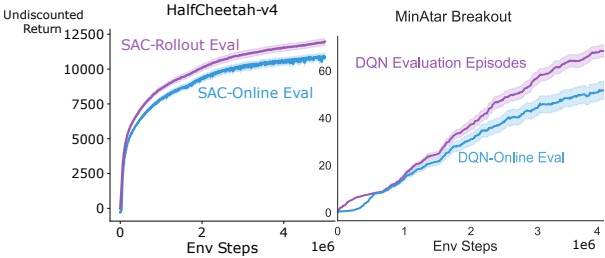

*Figure 4.* Contrasting the performance of SAC and DQN when using online evaluation versus evaluating deterministic policy in evaluation episodes. The solid lines are the average across 30 runs for SAC and 50 runs for DQN and the shaded region is the standard error.

Most RL papers, however, use evaluation rollouts, including in most papers using SAC (Haarnoja et al., 2018), DQN (Mnih et al., 2015), DDPG (Lillicrap et al., 2019), PQN (Gallici et al., 2025), Dreamer (Hafner et al., 2025), MuZero (Schrittwieser et al., 2019), and TD3 (Fujimoto et al., 2018). Evaluation rollouts are often used with the greedy policy, without exploration. These rollouts are likely

---

[3]There are potentially use cases for the simulator-as-a-proxy setting where an RL researcher might want to know: if at any point the agent were to stop learning and deploy it's fixed policy, then how good would it be? Such a setting would need to be clearly defined, as it implies the cost incurred in deployment is not the key criteria. When solving a simulator, on the other hand, evaluation rollouts are sensible as the agent is trying to identify when it has learned a good fixed policy.

to report higher performance than the online return, where the policy has exploration. Figure 4 shows how these two metrics can look different for SAC and DQN in two different environments. The evaluation episode returns in purple are much smoother and above the blue line showing the online performance. More details on this experiment can be found in Appendix A.4. The gap in this setting is actually not that big, but the results are nonetheless noticeably different. There are other settings where this gap would be much larger, such as when exploratory actions can dramatically derail the agent.

## 4. Call to Action

Our call to action is simple: RL researchers should clearly state their problem setting, and then stay within the appropriate restrictions when running experiments in simulators. In practice making this distinction would consist of a single declaration of intent, for example an author would state they are trying to solve a protein docking simulator. Once the intention is made clear researchers should remain consistent and adhere to the constraints their goals imply. For example, an empirical RL researcher attempting to understand online RL algorithms learning in deployment would likely use a single stream setting (no parallel environments) and assess performance during learning. An algorithm built with the goal of solving simulators will not be comparable to those concerned with learning in deployment. Following our call to action would foster research directed towards the goals each individual researcher cares about.

We can also try to recognize and avoid the reasons for why this confusion arose in the first place. One potential reason might be the prevalent usage of simulators in RL research, implying that a focus is on solving simulators and solving benchmarks. One call to action could be for the community to better reward papers that apply RL algorithms to their intended use case. A concrete step would be for our top venues to be more receptive of applications of RL in real physical systems, as has also been advocated for more generally in machine learning (Rolnick et al., 2024).

Another reason for the confusion might be the development of RL algorithms that work well in multiple settings. For example, we can use PPO with vectorized environments to efficiently solve a simulator (Weng et al., 2022), and we can also use PPO with a single environment to run on robots (Mahmood et al., 2018). Clearly such generality is useful. At the same time, however, these are quite different versions of PPO, to the point that one could consider them different algorithms. One additional call to action is for the community to consider explicitly labelling different version of algorithms (e.g., PPO-v1 and PPO-MultiEnv), to highlight the difference and remain clear about the problem setting.

A third potential reason is simply the allure of getting empirical results faster through the use of parallel environments. Restricting the use of the simulator, to be more like deployment, ends up hampering the experimenter themselves. For example, consider a physical system where actions are only taken every 5 minutes. The algorithms can use a lot of compute between steps, for example by significantly increasing the number of replay updates (the update-to-data ratio). Testing such algorithms, however, significantly slows down what could previously have been a fast experiment when there is only one update per step. Such a restriction may be necessary to properly assess the desired setting, but can also be painful, especially when there is a tendency to compare to the standard setting that gets more environment interaction. One call to action is to remain hyper-vigilant about the use of the simulator as a proxy. Another possible call to action is to consider novel empirical approaches that allow the use of parallel environments without compromising empirical conclusions. While our call-to-action is focused on reinforcement learning, we note that other communities using research simulators would benefit from a similar discussion. We invite such communities to explore how similar constraints may apply to their settings.

## 5. Alternative Views

This section outlines possible alternative perspectives to our position that researchers should distinguish between simulator usages.

**The simulator usage is always clear from the context.** There are cases where the intended goal of simulator usage is indeed clear. For example, it is clear that Degrave et al. (2022a) used the tokamak simulator to understand that specific application, rather than as simply as proxy to test and develop algorithms for learning in deployment. However, for simulators such as ALE (Bellemare et al., 2013), the intended goal varies and is often unclear. While several works used the ALE as a benchmark for measuring the competency of RL agents, others aim to solve it directly as a simulator (Lipovetzky et al., 2015; Jinnai & Fukunaga, 2017). Moreover, even when researchers reiterate ALE's original purpose—as a benchmark for general RL performance—they sometimes adopt evaluation metrics that misalign with this goal, such as evaluating the best-performing checkpoint (Mnih et al., 2015).

Let us consider another example of this ambiguity with Dreamer (Hafner et al., 2025), which achieved state-of-the-art results in Minecraft. Dreamer goes to great lengths to learn a useful model for planning. This choice suggests that Dreamer is using Minecraft as a proxy for learning in deployment, because otherwise it could plan with the simulator itself for greater accuracy. Furthermore, the authors suggest "this ability could also help to create robots that can learn to interact in the real world" (Biever, 2025). However, Dreamer uses 64 copies of the Minecraft simulator, which would suggest it is solving Minecraft, as discussed in Section 3.1.

**The solving simulators use case encompasses sim2real.** There are many instances of simulators that are solved for their own sake, with no intention to do sim2real. Examples include many search problems, like discovering protein docking molecules (Kim et al., 2024), material discovery for carbon capture (Al-Sakkaria et al., 2024) and discovering matrix multiplication algorithm using AlphaTensor (Fawzi et al., 2022). The resulting solutions may only be used for scientific understanding, even if at some later date some version of the found solution is used in deployment.

This position paper, however, does have bearing on sim2real. In the simulation phase, RL algorithms may be used to learn the initial policy. If the RL algorithm does not sufficiently exploit that it is solving a simulator, then performance may be left on the table, as discussed in section 3.1 and 3.2. If RL researchers distinguished their use of simulators, then sim2real practitioners could make more informed decisions on which algorithms to use for this first phase. The lack of distinction also affects the second phase, of deploying in the real world, since many RL algorithms inadvertently exploit simulators. For sim2real practitioners, this might mean using a SOTA RL algorithm for real world learning will perform worse than is reflected by results in the RL literature, as discussed in section 3.3 and 3.4. Distinguishing between simulator usages would ensure RL algorithms are more suited for both phases in sim2real.

**Research should be unconstrained to maximize output, and gaps between problem settings can be addressed later.** Another potential perspective is that the community should adopt whatever provides the fastest feedback loop for research in the hopes of accelerating progress. Accordingly the gaps highlighted throughout this work could be addressed at a later-date. Some of this future work would take the form of transferring ideas from the general literature to a particular setting, e.g learning in deployment. Even when adopting this particular alternative view, communicating the problem setting is still essential, and is a key aspect of our call-to-action. It would still be essential to be transparent about what aspect of a particular work relies on simulators, to facilitate future research that attempts to borrow any ideas for the learning in deployment setting. This alternative view, therefore, is not really counter to our position, as researchers are welcome to accelerate progress by focusing on work in simulation while also being transparent on current limitations to other settings.

**It is obvious that RL performs worse in deployment than in simulation so there is no need for the distinction between these settings to be discussed.** It is typically true

that learning in deployment, under real world constraints, is more challenging than learning in simulation. There are many reasons for this; real-time computational barriers, poor sample complexity of modern RL algorithms, and violation of stationarity assumptions, among others. Notably these problems can be solved or mitigated by the further development of RL algorithms. For instance, better recurrent learning algorithms can overcome partial observability in the real world. These deployment challenges, however, are different from the problems we highlight in this paper. No amount of algorithmic development will allow an agent to at will produce new copies of the environment in deployment. Our distinction must be made precisely so that advances in empirical RL research can benefit learning in deployment situations. We hope that if more work is focused on using simulators as a proxy for learning in deployment, it will encourage the use of simulators which mimic the aforementioned challenges of learning in deployment.

**The focus of RL is on the process of learning in deployment, and any technique which exploits simulators should be considered poor empirical design.** Some researchers may choose only to focus on learning processes which mimic learning in deployment. While this is an important research direction, it is not the only direction. RL has shown immense value by solving critical problems using high fidelity simulators, for example computer chip design (Mirhoseini et al., 2021). Similarly, RL has discovered novel strategies and moves through solving simulators of games such as Go (Silver et al., 2016) and Poker (Moravčík et al., 2017). There is no reason to reject these solution techniques and problems simply because they are not directly transferable to learning in deployment.

## 6. Conclusion

The position put forth in this paper is that RL researchers need to distinguish between solving simulators and using simulators as a proxy for learning in deployment. We highlighted that when solving simulators, algorithms can exploit parallel copies of the environment and simulator-exclusive actions (like resetting), that are not appropriate in deployment. To use simulators a proxy for learning in deployment, algorithms and experiments need to use appropriate restrictions on what the algorithms can do and what evaluation metrics are used. Yet, RL papers are often not clear about which setting they are developing algorithms for, sometimes implying the goal is general purpose AI algorithms for learning in deployment but then using an offline/rollout evaluation. We highlighted four potential issues and misleading conclusions that arise from not making the simulator use-case clear, including 1) highly different performance when being forced to use one copy of the environment rather than multiple, 2) suboptimal performance when not exploit-

ing simulator-exclusive actions like resetting, 3) issues with standard hyperparameter tuning strategies when using the simulator as a proxy for the real world and 4) differing conclusions when using online return versus evaluation rollouts for the simulator-as-a-proxy setting.

This paper advocated for clarity in these two specific uses of simulators, to make a focused argument. Of course, more generally an argument can be made for clarity of problem setting for other uses of simulators, such as in sim2real and using simulators as a proxy for solving simulators. The ultimate outcome if we improve on this clarity for all use cases is less confusion for when and how to use RL algorithms, especially for practitioners outside of the RL community, and hopefully more impact for all of our research.

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

# A. Experiment Details

Relevant code for experiments is provided at https://github.com/Matthew-Vandergrift/simulators_and_deployment.

## A.1. Collapsing Learning Curve Experiment

For the experiment which produced Figure 1 we ran PPO (Schulman et al., 2017) for fifty million steps, over 20 seeds. We reported the online return which was the curve that collapsed. We also reported the performance of the best policy found so far, ran in an evaluation episode. We started this after the timestep by which each seed had reached its best policy. This curve stayed at a constant performance. We used the hyperparameters shown in Table 3. The hyperparameters used are the same hyperparameters as (Dohare et al., 2023).

## A.2. PQN Experiment

To produce figure 2 we first ran PQN using the default hyperparameters as specified in the configuration files on its official Github page (Gallici et al., 2025). We then changed only the number of environments to 1 within the configuration to produce the default 1 environment result. Lastly we set the number of steps to 4096 with 1 environment. We did this because we reasoned that this would ensure each update step occurs with the same number of transitions as the default configuration of PQN, we label this as matched update-to-data ratio PQN. For each configuration we ran for five million steps on the Asterix-Minatar environment (Young & Tian, 2019), for 30 seeds.

## A.3. Resetting Experiment

As discussed in section 3.2 we solve the simulator MountainCar-v0 by augmenting PPO with resets. To accomplish this every 500 steps we set a flag which upon the start of the next episode resets the agent to a random state. This state is drawn from the following distribution, position$\sim U[0.2, 0.8]$ and velocity$\sim U[-0.04, 0.04]$. These ranges were chosen because they are relatively close to the goal and hence informative for the agent. After 2-million steps in the environment, we stop resetting completely and let PPO run for $2-$million more steps. We compare this approach to PPO without resetting which is the default PPO. For both PPOs we use the same hyperparameters given in Table 2. For both algorithms we report the online return.

## A.4. Evaluation Rollouts Experiment

For this experiment we ran SAC and DQN using both online and evaluation rollouts/episodes. For SAC our evaluation episodes consisted of deterministic action selection rather

| Hyperparameter Name | Value |
|---|---|
| Observation Normalization | False |
| Reward Normalization | False |
| Advantage Normalization | True |
| Gradient Clipping | True |
| Value Loss Clipping | True |
| Rollout Length | 2048 |
| Epochs | 4 |
| Minibatch size | 256 |
| $\lambda$ | 0.95 |
| Discount factor, $\gamma$ | 0.99 |
| Clip Coefficient | 0.2 |
| Optimizer | Adam |
| Learning rate | 0.0009 |
| Optimizer $\epsilon$ | $1e{-}5$ |

*Table 2.* PPO hyperparameters used for the experiments in Figure 3.

than sampling. We used the Half-Cheetah MuJoCo environment (Todorov et al., 2012). We run for five million steps over 30 seeds, plotting both the evaluation episode returns and the online return. The hyperparameters used for the SAC experiments in Figure 4 are shown in Table 4. For DQN we ran evaluation episodes which consists of greedy action selection, i.e taking the action with the max Q value in the current state. We average the evaluation returns over 30 evaluation episodes. We used the MinAatar Breakout environment(Young & Tian, 2019) in Gymnax (Lange, 2022), running for $4,000,000$ steps, plotting the evaluation and online return. The hyperparameters used are listed in table 5.

| Hyperparameter Name | Value |
| --- | --- |
| Observation Normalization | False |
| Reward Normalization | True |
| Advantage Normalization | True |
| Gradient Clipping | False |
| Value Loss Clipping | False |
| Rollout Length | 2048 |
| Epochs | 10 |
| Minibatch size | 256 |
| $\lambda$ | 0.95 |
| Discount factor, $\gamma$ | 0.99 |
| Clip Coefficient | 0.2 |
| Optimizer | Adam |
| Learning rate | 0.0003 |
| Optimizer $\epsilon$ | $1e{-}8$ |

*Table 3.* PPO hyperparameters used for the experiments in Figure 1

.

| Hyperparameter Name | Value |
| --- | --- |
| Epsilon Start | 1.0 |
| Epsilon Finish | 0.01 |
| Time to Anneal Epsilon | 400,000 |
| Target Update Interval | 500 |
| Batch size | 128 |
| Buffer size | 1,000,000 |
| gamma,$\gamma$ | 0.99 |
| tau, $\tau$ | 1.0 |
| LR Decay | False |
| Learning rate | 0.00025 |
| Evaluation length | 30 episodes |
| Evaluation frequency | 500 steps |
| Steps before learning | 10,000 |
| Total Steps | 4,000,000 |
| Optimizer | Adam |

*Table 5.* DQN hyperparameters used for the experiments in Figure 4.

| Hyperparameter Name | Value |
| --- | --- |
| Batch size | 256 |
| Buffer size | 1,000,000 |
| gamma,$\gamma$ | 0.99 |
| tau, $\tau$ | 0.005 |
| Number of exploration steps | 10,000 |
| Learning rate | 0.0003 |
| Evaluation length | 10 episodes |
| Evaluation frequency | 4,000 steps |

*Table 4.* SAC hyperparameters used for the experiments in Figure 4.

