# OpenReview forum: "Position: RL Researchers Need to Distinguish Between Solving Simulators and Using Simulators as a Proxy"
_ICML.cc/2026/Position_Paper_Track — ICML 2026 Position Paper Track regular_

### Official Review · Reviewer_TmK1 · 2026-02-22

**Significance:** 2
**Argument Clarity:** 3
**Rating:** 4
**Confidence:** 2

**Questions:**

Please see the questions in the weaknesses.

**Alternative Views Section:**

Yes

**Compliance With Llm Reviewing Policy A Conservative:**

Affirmed.

**Discussion Potential:**

2

**Final Justification:**

This paper presents a timely and important critique of a fundamental methodological ambiguity in reinforcement learning (RL) research. The central topic that RL researchers must clearly distinguish between "solving simulators" and "using simulators as a proxy for learning in deployment" is well-argued and supported with concrete examples and experiments. The paper effectively diagnoses a source of confusion that has likely led to misaligned expectations and slowed progress in translating RL research to real-world applications. Also, the rebuttal has addressed my comments. Therefore, I raised my score.

**Paper Summary:**

This position paper proposes a call that RL researchers need to distinguish between solving simulators and using simulators as a proxy. The authors discuss how these two use-cases are importantly different, in terms of constraints on how the agent can use the simulator, which algorithms are appropriate and which evaluation metrics are appropriate. They then highlight several issues and misleading conclusions
that can occur by not making the distinction between these two settings clear, supported with examples and simple experiments. The paper is clearly written, well-structured, and its argument is bolstered by illustrative experiments that make the consequences of this confusion tangible.

**Position:**

Yes

**Position In Title:**

Yes

**Related Work:**

3

**Strengths And Weaknesses:**

Strengths:

1. The paper identifies a genuine and widespread issue, which is the distinction between "solving a simulator" and "using a simulator as a proxy".

2. The paper provides a clear comparison of the two cases across interaction options, solution techniques, and evaluation metrics.

3. The illustration of "issues" in this work is compelling and extensive.

4. Section 5 does a good job of anticipating and responding to potential counterarguments.

5. Also, the paper doesn't just critique, and it offers a simple and practical solution, which is to clearly state the problem setting and staying within the appropriate restrictions. The suggestions for labeling algorithm variants (e.g., PPO-v1, PPO-MultiEnv) and rewarding real-world applications are concrete steps the community can take.

Weaknesses:

While we have seen the technically solid points from the paper, there are still a few areas for improvement and critical questions.

1. The paper presents a sharp dichotomy, but acknowledges that algorithms like PPO can be used in both settings. The paper's solution is to label them as different algorithms (PPO-v1, PPO-MultiEnv). However, this might be an over-correction. Many algorithm designs are intended to be general, and the core innovation might be orthogonal to the number of environments. Forcing a split could obscure the fact that a single algorithmic insight (e.g., a new loss function) works well regardless of the setting. The paper could explore this nuance more deeply. Is the problem the algorithms themselves, or the experimental protocols and claims made about them?

2. The paper mentions that using a single environment to emulate deployment can be "painful" and "slow down" experiments. This is a real and significant barrier. If the community adopts this distinction, how do we make research in the "proxy" setting tractable? Are there ways to use parallel resources for hyperparameter tuning or ablation studies without contaminating the core results? The paper briefly mentions "novel empirical approaches," but this could be a whole research agenda in itself. For example, could one run a fast and massively parallel hyperparameter sweep to identify promising candidates, and then do a final and definitive evaluation in a single-stream setup? How would this be reported to avoid the issues raised in Section 3.3?

3. The paper's "proxy" setting is implicitly defined by what it cannot do (no parallelism, no resets, no cost-free tuning). However, it might be beneficial to define it more positively. What are the key features of a real-world deployment that a simulator-as-a-proxy should emulate? The paper mentions "sequential observations" and that "all rewards matter." But real deployment also has other constraints: real-time computation limits, partial observability, non-stationarity, and the potential for catastrophic failure. Should a good proxy attempt to emulate these as well?

4. Also, how is the "best policy" selected? If you are using a simulator, you can evaluate many candidate policies offline. This is a form of selection that itself exploits the simulator (as you can test policies without incurring cost in the "real" environment). The paper's framework implicitly allows this, but it's worth noting that even within the "solving a simulator" paradigm, there are further distinctions.

5. Though this work focuses on RL, the core issue is broader. This critique likely applies to any field that uses simulators for research, such as robotics, control theory, and even parts of supervised learning (e.g., using CIFAR-10 as a proxy for real-world image classification). A brief discussion of how this distinction might generalize could broaden the paper's impact.

6. Can the authors clarify the "Tuned PQN" in Figure 2? The caption says "tuned PQN" but the method section (A.2) describes it as running the same number of updates with 1 environment by increasing steps to 4096. Is this "tuning" or simply matching the update-to-data ratio? The term "tuned" might be slightly misleading. A clearer description in the main text would be helpful.

7. In Section 3.4, the text states "Most RL papers... use evaluation rollouts." The provided example (Figure 4) shows SAC and DQN. A citation to a survey or a stronger statement about the prevalence of this practice would be useful.

**Support:**

3

---

> ### Author Rebuttal · Authors · 2026-03-31
>
> We thank the reviewer for the valuable feedback. We address each point individually below.
> ### Weakness 1
> We discuss the sharp dichotomy concern under header Weakness 2 and 3 in our response to reviewer 6fch. We agree that there are cases that don’t fall into one of these categories. However, we believe that our call to action will also help those cases. For instance a researcher working on methods that benefit both settings can simply state this, and future researchers can adopt their method regardless of their setting. Right now, researchers do not clarify their settings, as a result research developments that benefit both settings might go unnoticed in one of them.
> ### Weakness 2
> We agree that this is a valid concern. Many techniques for solving simulators speed up research, and removing them may introduce challenges. However, many, but not all, of these techniques can be applied in a way which does not exploit the simulator. For instance instead of running one agent on parallel environments, running separate parallel agents to quickly get many independent seeds/trials will speed up results to some extent and is not exploiting the simulator.
>
> Hyperparameters are especially a hard problem for deployment. We agree that it requires a new research direction. However, concealing the problem is not a solution. If researchers keep the current practices that do not respect the deployment constraints, then there is little hope that RL algorithms will work well for such a setting. We believe that there is a need for new tuning algorithms that do not need resetting between tuning trials, something similar to sequential bayesian optimizers but adapted for deployment setting. Now, until these new algorithms are developed, researchers should report how they tune their hyperparameters, even if this tuning was done in parallel. A discussion on the limitations of their algorithms, and how they could extend their algorithm to the intended deployment setting should also be discussed.
> ### Weakness 3
> We defined the proxy setting in contrast to the simulator-solving setting, but we will rephrase it to be a more positive framing. We provide the new phrasing in our response to reviewer Tb1v. We also agree that the real world has other constraints. We believe that once our proposed distinction is widely adopted, there will be more incentive to study proxy simulators which further mimic challenges of the real world, e.g.  partial observability, and non-stationary. We will add a sentence in the conclusion about this as a future direction.
> ### Weakness 4
> We agree with the reviewer and our goal in figure 1 was exactly to show the contrast between the case where it’s possible to evaluate candidate policies and choose the best one versus the case where it’s not possible to do so. We weren’t allowing the use of best policy evaluation, rather contrasting to it and showing that the two evaluation methods result in different conclusions. We will further clarify this paragraph to explain that the best policy evaluation is only possible for the simulator case.
> ### Weakness 5
> We agree that the core issue exists in other domains as well. Robotics, and control theory are obviously similar as they involve sequential decision making, and the supervised settings where this most applies is in online supervised learning which shares similarities to the RL prediction problem. We will add a sentence in the conclusion inviting researchers from those other communities to investigate how similar constraints may apply to the specifics of their setting.
> ### Weakness 6
> There are two possibilities to run PQN with a single environment: 1) use all the hyperparameters from PQN but simply switch to a single environment. This results in doing the same number of updates as PQN but each update will have fewer samples. We referred to this as PQN with 1 Environment Default Hypers. 2) increase the number of steps between updates to 4096 so each update uses the same number of samples as the parallel environment version of PQN, but overall doing less updates as we have to wait to collect all the samples. We referred to this one as PQN with 1 Environment Tuned Hypers. The word ‘tuned’ here was simply to illustrate the update frequency change, which is a hyperparameter. We will change the name to “matched update-to-data ratio”.
> ### Weakness 7
> As far as we know, there are no surveys discussing the prevalence of evaluation rollouts, which is another reason why this paper is needed! Nonetheless, the use of evaluation rollouts exists in most RL papers. Major RL algorithms mentioned the use of evaluation rollouts including SAC, DQN, Rainbow, Dreamer, MuZero, TD3, and DDPG. The fact that all these algorithms used evaluation rollouts without further discussing the implications of such a choice confirms our position that the researchers are not distinguishing different settings and their constraints. We will add more citations to major RL works which use evaluation rollouts.

---

> > ### Author Rebuttal · Reviewer_TmK1 · 2026-04-02
> >
> > I really appreciate the rebuttal from the authors. Most of my concerns have been addressed. I hope the authors will incorporate the changes based on the comments into the revised draft . I will adjust my evaluation.

---

### Official Review · Reviewer_oZp8 · 2026-03-11

**Significance:** 3
**Argument Clarity:** 3
**Rating:** 5
**Confidence:** 4

**Questions:**

None.

**Alternative Views Section:**

Yes

**Compliance With Llm Reviewing Policy A Conservative:**

Affirmed.

**Discussion Potential:**

3

**Final Justification:**

The authors have answered my (minor) weaknesses and I keep recommending acceptance.

**Paper Summary:**

The authors argue that the RL community needs to clearly distinguish between
two scenarios in their work, which often get neglected and mixed up.
These are the distinction between the task of _solving a simulator_, i.e.,
achieving the best performance in a virtual scenario without relevance to real-world deployment,
vs. the opposite of _using a simulator as a proxy_, i.e., with the explicit goal of mimicking
real-world learning.

**Position:**

Yes

**Position In Title:**

Yes

**Related Work:**

3

**Strengths And Weaknesses:**

### Strengths
- The paper targets an important distinction that is often not highlighted explicitly, or even (un)intentionally obscured by vague promises of real-world significance in many RL papers. As such, the paper fits in well with a series of recent works, such as Patterson et al. (2024)'s work on how much we can even trust our empirical results within a simulator, and Castro (2025)'s observations on the discrepancy between theory and experimental implementation.
- The paper is well written, and the position is well argued, with empirical evidence provided for each of the raised issues.

### Weaknesses
- The formulation of most of the paper is rather binary, suggesting that these are the only two objectives for an RL researcher. A paper proposing a new method might rely on a broad set of simulators to evaluate how an agent learns (e.g., which exploration patterns). While better performance is often one of the proxy metrics-i.e., it technically falls into the first case-solving, for example, a MuJoCo benchmarking task would not be the goal of such a work; rather, it would again be the use of a simulator as a proxy, this time for a theoretical evaluation.


_____
Patterson et al. (2024), _Empirical Design in Reinforcement Learning_
Castro (2025), _The Formalism-Implementation Gap in Reinforcement Learning Research_

**Support:**

3

---

> ### Author Rebuttal · Authors · 2026-03-31
>
> We appreciate the reviewer’s insightful comments and suggestions. We address the weakness below.
>
> ### Weakness 1
>
> We agree with the reviewer that there are other objectives for an RL researcher, for instance sim2real which we briefly discuss in the paper.  We wanted to clarify that our position is not that these are the only two usages of a simulator, nor that they are the only two goals in RL. However, they constitute a substantial portion of RL research. We also believe that our call to action generalizes to settings that might not fall neatly into one of these categories. For example, consider a researcher working on better optimizers for RL. Their approaches can be applied to both methods for solving a simulator or methods for deployments. All the researcher needs to do is to clearly state this fact and explain how they chose their experimental setting.
>
> We will expand our remarks paragraph at the end of the introduction with more examples to further clarify other cases that don’t fall into one of these categories or maybe fall into both. One example we will include is of a researcher proposing a novel exploration strategy which may be applicable both in deployment and in simulation, compared to an exploration strategy which resets to random simulation states which cannot be applied to deployment.
>
> We also appreciate the positive comment from the reviewer about the thematic connection to the previous work of Patterson et al. (2024) and Castro (2025).  We will add a sentence relating our work to theirs as these works discuss complementary aspects of misalignments in empirical research practices.

---

> > ### Author Rebuttal · Reviewer_oZp8 · 2026-04-02
> >
> > Thank you for your rebuttal.

---

### Official Review · Reviewer_6fch · 2026-03-17

**Significance:** 3
**Argument Clarity:** 4
**Rating:** 5
**Confidence:** 4

**Questions:**

N/A

**Alternative Views Section:**

Yes

**Compliance With Llm Reviewing Policy A Conservative:**

Affirmed.

**Discussion Potential:**

4

**Final Justification:**

I maintain my recommendation to accept this paper. The rebuttal doesn't change my evaluation.

The authors present a relevant, well-reasoned argument that the RL community must explicitly distinguish between "solving a simulator" and "using a simulator as a proxy for deployment." When weighing the review dimensions, I think the paper's strengths outweigh its weaknesses. The conceptual framework is highly logical, and the empirical demonstrations (specifically showing how parallelized algorithms like PQN fail in single-stream environments) effectively highlight the pitfalls of current evaluation practices. Furthermore, the paper goes beyond criticism to offer actionable, constructive advice (e.g., the "declaration of intent" and algorithm versioning).

**Paper Summary:**

This paper argues that RL researchers must explicitly distinguish between two different use cases for simulators: solving a simulator and using a simulator as a proxy for learning in deployment. The authors argue that these two goals require different interaction constraints, algorithmic choices, and evaluation metrics.

The main contributions of this paper go as follows:

- It defines the constraints for each setting. "Solving a simulator" allows for parallel environments, simulator-exclusive actions like state-resetting, and evaluation based on the best policy found. Conversely, "using a simulator as a proxy" restricts the agent to a single interaction stream where rewards during learning are critical, and evaluation should rely on online returns.
- The paper also highlights some major issues arising from blurring these lines:
    1.  Impressive results from parallelized algorithms (e.g., PQN) do not translate to single-environment deployment.
    2.  Performance is "left on the table" when solving simulators because researchers often fail to exploit simulator-exclusive resets.
    3.  Standard hyperparameter tuning (erasing failed runs) exploits the simulator and ignores the real-world costs of configuration errors.
    4.  Evaluation rollouts mask the detrimental costs of exploration that would be present in a deployment setting.
- The authors urge researchers to provide a "declaration of intent" for their experiments and adhere strictly to the constraints of their chosen problem setting.

**Position:**

Yes

**Position In Title:**

Yes

**Related Work:**

3

**Strengths And Weaknesses:**

## Strengths

- The core argument is built on a clear logical foundation: if a simulator is a proxy for the real world, it must inherit the real world’s constraints. The authors provide a strong conceptual framework by identifying three distinct areas where the two goals diverge.


- The paper also provides experimental results that illustrate its points. For instance, Figure 2 demonstrates how an algorithm like PQN, which appears SOTA when parallelized, fails to maintain that performance when restricted to the single interaction stream typical of deployment settings.

- Rather than just criticizing current practices, the authors provide actionable advice, such as a "declaration of intent" and explicit versioning for algorithms (e.g., PPO-v1 vs. PPO-MultiEnv).

## Weaknesses

- While the paper effectively shows how "solving a simulator" is distinct, it provides less evidence that strictly following the "proxy" restrictions leads to more successful real-world deployments. The link between these empirical restrictions and final deployment success is largely theoretical.


- The authors explicitly exclude other significant RL use cases, such as sim2real, to maintain a manageable scope. This choice might lead some readers to feel the paper ignores the most common bridge between simulation and the real world.

- The paper presents a binary choice, but in practice, researchers often pursue both goals simultaneously (e.g., finding a fast solution to a problem while hoping the algorithm is general-purpose). The paper could benefit from addressing how to handle research that spans both categories.

**Support:**

3

---

> ### Author Rebuttal · Authors · 2026-03-31
>
> We thank the reviewer for their supportive comments and their advice regarding avoiding an overly binary framing. We will address the weaknesses listed point by point below.
>
> ### Weakness 1
>
> We thank the reviewer for considering how our position would impact real-world deployments. We showed several examples where the current practices do not lead to success when the constraints of the deployment setting are enforced. We believe that this demonstrates the problem, and once this problem is exposed further research is definitely needed to propose new methodologies that guarantee successful transition from research to deployment. However, our goal from the position paper is to simply expose the problem and we believe our call to action will ignite new research directions studying this problem further.
>
> ### Weakness 2 and 3
>
> We address both the weaknesses about overall binary framing and the lack of focus on other use cases. We acknowledge that there are some research questions that might not fall into one of these two categories, such as sim2real which we briefly discuss in the paper. We wanted to clarify that our position is not that these are the only two usages of a simulator, nor that they are the only two goals in RL. However, they constitute a substantial portion of RL research.
>
> We also believe that our call to action generalizes to settings that might not fall neatly in one of these categories. For example, consider a researcher working on better optimizers for RL. Their approaches can be applied to both methods for solving a simulator or methods for deployments. All the researcher needs to do is to clearly state this fact and explain how they chose their experimental setting. We will expand our remarks paragraph at the end of the introduction with more examples to further clarify other cases that don’t fall into one of these categories.
>
> We thank the reviewer for their feedback and hope we have addressed their concerns to improve our paper.

---

> > ### Author Rebuttal · Reviewer_6fch · 2026-04-03
> >
> > Thanks for the response. I still recommend acceptance.

---

### Official Review · Reviewer_Tb1v · 2026-03-20

**Significance:** 4
**Argument Clarity:** 4
**Rating:** 6
**Confidence:** 4

**Questions:**

1) *What is a simulator and what is deployment?* I think another iteration could be made in formally defining what is a simulator and what is deployment in this context. The paper states the difference between "simulator" and "proxy" as "the ability to use parallel environments and the availability of simulator-exclusive actions". However, one can argue that some *deployment* domains also have those features, e.g., a web agent interacting with parallel instances of a web browser, although actions (credit card transactions, bookings) may have a tangible impact in the real world (is this simulator or deployment?) Another instance is a factory with multiple robots operating in parallel: While this interaction happens on a physical system, the RL algorithm may receive data asynchronously from all the robots. I do not see this as a major issue, but perhaps an indication that the categorization is not clear-cut.

2) *Samples vs compute frequency.* One aspect that I believe crucially distinguish "learning in simulation" and "learning during deployment" is oftentimes the frequency of computation against the frequency of sampling. In RL simulators, it is often common to sample many times for a single update (let's say a single *computation*), whereas, in practice, the sampling can be very slow, so that a lot of computation may be carried out between one sample and the next (e.g., AlphaGo waiting for the other player move can take a lot of preliminary computation). This is hinted at in the Alternative Views section, but perhaps could be highlighted previously in the paper as a core difference?

3) *Alternative views.* Some ideas of alternative views that could be included.
- Stating the setting clearly: Perhaps the categorization between "solving simulator" and "simulator as a proxy" does not capture the intricacies of the settings considered in the RL research, for various reason. In this view, the call for action may be rephrased into stating the setting clearly and asking the authors to include a specific section that clarifies how the proposed algorithm may be implemented in the application it is intended for ("in deployment").
- Unconstrained research: Another potential counter-argument is that, as a community, we shall put as little constraints as we can on the research production, so that to have the fastest feedback loop between ideas-test. While this may cause some drift from "deployment" settings, a subsequent effort could be made to retain and highlight the ideas of practical relevance. On the other hand, prioritizing research on "simulator as a proxy" in ML conferences may slow down the ability of the community to produce progress.

Minor comments:
- Is it really parallel computation the problem or rather asynchronous updates? If the updates are deferred, one can argue that parallel computation is equivalent to sequential computation, especially if reset is naturally available (episodic settings).
- Figure 2: "A global step is one step in each available simulator". It is not fully clear what does this mean, but I think the standard practice is to show the performance against the total number of collected samples (parallel or not).
- Figure 1: I am not sure the "best policy evaluation" curve is discussed.
- Hyper-parameter tuning: I think tuning the hyper-parameters does not only extract the most from a proposed algorithm, but also allows for a somewhat fair comparison with baselines (results of two algorithm *with their best parameters*). Avoiding the usual ways for hyper-parameter tuning could harm the fairness of algorithms comparisons. Moreover, in some deployment settings, parameter tuning is not that different than grid search, e.g., A/B testing.

**Alternative Views Section:**

Yes

**Compliance With Llm Reviewing Policy A Conservative:**

Affirmed.

**Discussion Potential:**

4

**Final Justification:**

I thank the authors for their thorough reply to my comments. I think some of their replies can be incorporated in the paper to make it even more clear and sharp. I maintain my fully positive score about the paper.

**Paper Summary:**

The paper position is about the need for RL papers to explicitly state whether the intended goal is to solve a simulator or a simulator is solely used as a proxy of the intended real-world environment. Especially, the paper distinguishes between "solving a simulator", when data comes from parallel instances of the environment, the state of the process can be reset, multiple outcomes can be generated from a state, and "using a simulator as a proxy", where none of the latter options are available. The paper argues that the two settings shall be dealt differently in terms of algorithmic solutions and evaluation (e.g., planning and grid-search hyperparameter tuning can be used in "solving a simulator", but not in "simulator as a proxy"). Thus, RL papers shall explicitly state which of the two settings are targeting.

**Position:**

Yes

**Position In Title:**

Yes

**Related Work:**

4

**Strengths And Weaknesses:**

Strengths
- The position expressed by the paper is relevant and important for the RL community, especially empirical research;
- The position is well described, supported, and thoughtful;
- The position is likely to spark a positive discussion on how to experiment RL algorithms at the conference;
- The position states a call for action that is reasonable and actionable.

Weaknesses
- The paper does not formally define what is a simulator vs a deployment setting. One can argue that some "deployment" settings actually have some of the features of "solving a simulator".
- The "alternative views" section is somewhat weak. On the one hand, I believe this is due to the position being very reasonable, so that counter-arguments are not obvious. However, the section itself could be dedicated more to support the alternative views rather than to explain why they are limited.

Evaluation

I believe the position expressed by the paper is very important for the RL community. I fully recommend accepting the paper, and possibly highlighting it at the conference. I think the paper could be further improved (some comments below), but none of those aspects justify reject in my view.

**Support:**

3

---

> ### Author Rebuttal · Authors · 2026-03-31
>
> We thank the reviewer for positive feedback about the importance of this issue for the RL community, as well as insightful questions and suggestions. We address the questions point by point below.
>
> ### Question 1
> We agree that there are some cases where the separation is not as clear, or there might be overlap in the properties of the two usages. We tried to clarify that it is not always a clear cut with the sim2real discussion at the end of the introduction, but we plan to further clarify this with further examples. The example of an agent interacting with several instances of a web browser is an interesting one as a deployment scenario that has similar properties to solving a simulator.
> In terms of formally defining a simulator, we would state that: A simulator is software implementation of specified dynamics in which interactions are consequence-free: transitions and rewards are recorded but not physically enacted, so that no real-world costs, risks, or irreversible effects are incurred. Furthermore, since actions are not physically enacted; parallel copies, resetting to arbitrary states, and querying multiple outcomes from the same state-action pair are possible regardless of if they are intended in the specified dynamics. This definition addresses some of the lack of clarity about our categorization. Our deployment definition is the scenario where interactions are enacted and matter, which is an alternative setting. We will include such a definition for improved clarity in our position.
>
> ### Question 2
>
> This is an interesting point we appreciate the reviewers perspective on. The reason we choose not to highlight this is that the sampling frequency can be varied across deployment environments. For instance there are fast deployment environments such as robotic locomotion, and slow environments such as industrial control or playing go. For us the distinction is that in a simulation environment, the simulator can be manipulated to avoid dealing with an undesired sampling frequency either by slowing it down or running parallel queries for more data, whereas in deployment this frequency is fixed and must be dealt with.
>
> ### Question 3
>
> For the first proposed alternative view, we tried to make our call to action simple to reduce the barrier to implementing it. However, we agree that researchers should discuss how their proposed methods could be implemented for an intended application. We will include this point in our call to action. We similarly plan to incorporate the second alternative view about unconstrained research. In this section we will also discuss the downside of this view, which is that it will increase the gap between research and deployment, and reduce the applicability of RL algorithms in real-world scenarios.
>
> ### Minor Comment 1
>
> It’s true that most usages of parallel environments come with asynchronous updates such as in A2C or IMPALA. However, there are cases where researchers use parallel computation and updates are synchronous, such as in PPO and PQN. We chose to frame the problem in terms of parallel environments to avoid confusion between these two. We used the term parallel since the word asynchronous is used in multiple contexts, such as asynchronous updating in robotics.
>
> ### Minor Comment 2
>
> We wanted to clarify the difference between two approaches for counting the steps. Approach 1 has one step corresponding to one sample, so using n parallel environments increments the step by n each time. Approach 2 has one step corresponding to one call to env.step() so the n parallel environments are counted as one step. The reason for this distinction is that when people report wall time calculations, they are implicitly doing approach 2 which conceals the fact that they are using more samples. PQN reports that their algorithm is fast for this exact reason.
>
> ### Minor Comment 3
>
> We will clarify in the main text that the best policy evaluation was a saved checkpoint from the policy that had the highest return before the dashed line.
>
> ### Minor Comment 4
>
> We totally agree! There are more issues to hyperparameters that would expand the scope of the paper. Even reporting the results with the best hyperparameters is not always fair; an algorithm with more hyperparameters or highly sensitive to the choice of its hyperparameters will use more samples to tune its hyperparameters than an algorithm with a fewer number of hyperparameters. Using more samples matters in deployment, but can be concealed if done in a simulator without consequence. We don’t necessarily advocate for avoiding tuning, but for respecting the constraints of the problem setting. We believe that once the constraints are clear and enforced, researchers will work on better tuning algorithms. For example, one direction could be to further work on sequential tuning algorithms that don't require environment reset between tuning trials, and thus are more suitable for real-world deployment.

---

> > ### Author Rebuttal · Reviewer_Tb1v · 2026-04-01
> >
> > I thank the authors for their thorough reply to my comments. I think some of their replies can be incorporated in the paper to make it even more clear and sharp. I maintain my fully positive score about the paper.

---

### Decision · Program_Chairs · 2026-04-30

**Decision:**

Accept (regular)

**Comment:**

The paper argues that RL researchers should distinguish between “solving a simulator” and “using a simulator as a proxy for deployment,” and supports this position with clear conceptual arguments and illustrative experiments. All reviewers agree that this is a timely and important methodological issue, and that the paper is well written, well argued, and likely to stimulate valuable discussion. The actionable recommendations (e.g., clarifying experimental intent and reporting practices) are a particular strength.

The main concerns relate to framing: several reviewers noted that the distinction can appear overly binary, and asked for clearer definitions and a stronger discussion of ambiguous cases and alternative views. However, these are largely issues of exposition rather than substance. The authors’ rebuttal was constructive, and all reviewers maintained their positive recommendations.

Overall, the paper makes a solid and relevant contribution to methodological discussions in RL, and its strengths outweigh its weaknesses.